# How to Make an Internal Team Coach: An Integration of Research

**DOI:** 10.3390/bs14060452

**Published:** 2024-05-27

**Authors:** Gabriela Fernández Castillo, Rylee Linhardt, Eduardo Salas

**Affiliations:** Department of Psychological Sciences-MS 25, Rice University, 6100 Main St., Sewall Hall, Houston, TX 77005, USA; rml5@rice.edu (R.L.); es32@rice.edu (E.S.)

**Keywords:** team coaching, leadership training, team training

## Abstract

Team coaching has been found to increase group effort, improve interpersonal processes, and increase team knowledge and learning. However, the team coaching literature is renowned for its inability to define team coaching itself—making it difficult to solidify its place in the world of team science. So far, there is no consensus on what specific training would serve internal leaders best, and how they would connect to the team coaching literature. We know leadership and team training are effective in improving organizational outcomes, but the gap in the literature lies in identifying what specific competencies internal team coaches need, and what training could fulfill these. In this piece, we seek to (1) identify what competencies internal team leaders need based on the outcomes we know team coaching yields, (2) identify specific behaviors that can fulfill these competencies, and (3) integrate the literature to form an evidence-based guide on what training to provide to internal team coaches. By doing so, we hope to provide a definitive understanding of what internal team coaches need to be successful.

## 1. Introduction

Collaboration demands have surged by over 50% in recent years [1]—underpinning a clear need for effective teamwork. However, as organizations continue to require and seek out “team players”, many have erred in assuming that teamwork is an inherent skill [2]. On top of this erroneous belief, many individuals who lead teams are also promoted or hired on the assumption that their past task performance translates to effective *leadership* performance [3]. With this, a conglomerate of issues can arise, as team members do not know how to act as a team, and team leaders do not know how to lead one. These misconceptions have paved the way for executive coaching. Executive coaching uses questioning to help clients unlock their professional potential and has gained substantial traction in the organizational world, generating nearly USD 3 billion in profit in 2019 [4]. Profit aside, executive coaching is widely recognized as an effective intervention that yields a variety of positive work outcomes, such as increased work satisfaction, better time management, and increased employee productivity [5,6]. Notwithstanding, its “sister-like” approach, *team coaching*, has been largely underutilized and underdeveloped [7].

*Leadership* can be understood as a process of social influence that involves directing people towards a common purpose [8] (p. 68). Leaders enact leadership in a variety of ways, from laissez-faire hands-off approaches to more involved ways, such as team coaching. Team coaching can be understood as a way of leading that emphasizes “direct interaction with a team intended to help members in the coordinated and task-appropriate use of their collective resources in accomplishing the team’s work” [9] (p. 269). Until recently, team coaching had been known as a “black box” [10], where team performance improved, but it was difficult to determine why. Recent advances have integrated team coaching work and determined that there are three empirically supported ways in which team coaching works [2]: it improves group effort/motivation (i.e., [10]), it improves interpersonal processes via increases in psychological safety (i.e., [11]), and lastly, it improves team knowledge and learning (i.e., [12]). Altogether, these three processes can improve team performance [2].

As conceptualized by Hackman and Wageman [9] and supported by the literature, team coaching can be effectively carried out by an internal team leader [13]. This approach differs from executive coaching, where typically, an external consultant is hired to develop one individual at a time [14]. The benefits of using an *internal team leader* as an *internal team coach* (known as the leader-behavioral approach) are particularly valuable for certain teams, such as those who have well-defined boundaries (e.g., the task is clearly defined) and teams that are more “compact” (e.g., on the smaller side, 4~10 members). The use of internal team coaches can eliminate the need for later external intervention. On top of this, internal coaches can bring a wealth of task-related knowledge that makes them highly attuned to the team’s dynamic [2]. For example, a surgeon who operates as an internal team coach knows their team’s weaknesses and strengths simply by working with them rather than having to observe them, and they understand the technicalities involved in the shared goal of a successful surgical procedure. See Table 1 for clarifying definitions.

In comparison to executive coaching, team coaching has been slower to “take-off.” Though there are a variety of factors at play, the literature has long grappled with a lack of clarity in defining team coaching [7] and having a consistent approach to implementing the intervention [15]. Until recently, this heterogeneity made it difficult to determine what outcomes team coaching can and cannot yield [2]. Yet, on top of this, team coaching has not been extensively studied empirically, with some exceptions (e.g., [13]). Given that external coaches usually undergo extensive certifications and training to perform their work [14], it is difficult to recommend the use of internal team coaches without first understanding what competencies they should hold and what tasks they should execute. Therefore, to continue strengthening the team coaching literature, this paper seeks to identify what competencies internal team coaches need based on the outcomes we know team coaching yields—from increasing group effort to improving interpersonal processes to improving team knowledge and learning. By doing so, we answer a call to understand the core competencies of a team coach [16].

It is important to state that there are a variety of existing team coaching models outside of the Hackman and Wageman [9] framework—such as the Systemic Team Coaching model [17] or Thorton’s [18] psychodynamic perspective. We proceed in using the Hackman and Wageman [9] internal approach alongside its empirically corroborated propositions [2] not only because there is simply more evidence behind the leader-behavioral approach [19], but also because this framework is task-focused and useful for teams that have certain characteristics, such as a clear task and a compact membership structure. We do not negate the strengths of other frameworks, and in fact, this review purposefully includes a plethora of team coaching work, from external to internal approaches to organize it under this overarching framework. We hope that by identifying the key competencies of an effective team coach, we provide homogeneity in the team coaching as a leader-behavioral approach whilst incorporating a variety of perspectives and advances made in recent years. Hence, our goal is to take team coaching’s diversity and organize under this framework.

## 2. An Integration of Leadership, Team Training, and Team Coaching Research

To our knowledge, there is no set standard for how to train an internal team coach. The discrepancies in defining the approach have made it difficult to move forward, and the significant lack of empirical work greatly constricts the conclusions we can reach. Nevertheless, the leader-behavioral approach is well positioned because it inevitably involves three streams of research: leadership, team science, and team coaching literature. The leadership training literature has decades of empirical support and a plethora of meta-analytic evidence suggesting that leadership training works (see [20]). The team training literature is also heavily supported by empirical (e.g., [21]) and meta-analytic evidence (e.g., [22]). Moreover, the team coaching literature also contains important insights that use a variety of frameworks outside of the Hackman and Wageman [9] approach. Given that the team coaching literature is nascent compared to the other two, we undertake an integrative approach, where we systematically review these three streams of literature to put forward the core competencies an effective internal team coach should have. In this review, we define a competency in alignment with prior team research, which are the knowledge, skills, and abilities (KSAs) “critical for team [leaders] to interdependently interact with [their team] in such a way that leads to positive team-based outcomes” [20] (p. 518).

Nevertheless, it is important to understand an assumption of this methodology is the idea that what would work in a leadership training program, a team training program, or another type of team coaching program (e.g., external approach) would apply to an internal team coach. We recognize the limitations of this assumption—and encourage further empirical work to validate the below competencies—but given the lack of existing research, we move forward in using prior research to inform this new approach. By doing so, this research also presents the strength that the competencies provided represent overlap between all three streams of literature, speaking to their potential generalizability.

## 3. Methodology

### 3.1. Search Strategy

This systematic review was performed according to the Preferred Reporting Items for Systematic Reviews and Meta-Analyses (PRISMA) guidelines [23]. Three keywords were utilized: “Leadership training” or “Team coaching” or “Team training”, and three databases were searched: Academic Search Complete, PsycInfo, and Business Source Complete.

### 3.2. Exclusion and Inclusion Criteria

The initial search yielded 12,144 articles. The authors automatically removed any articles that were not peer-reviewed, in English, in an academic journal, and whose full text was not available through the university database. Once duplicates were automatically removed, this yielded a total of 1821 articles (see Figure 1). We then excluded articles that included announcements, commentaries, corrigenda, or articles that included information about mental health, substance abuse, family law, minor information, spirituality, and disability training. We only assessed articles that contained information about leadership training, or team training, or team coaching, or coaching, and/or team/leadership competencies. Then, we proceeded to only include articles if they contained information about improving motivation/effort, improving interpersonal processes, or improving knowledge and learning, and/or general team/leadership competencies.

### 3.3. Consolidation of Themes

As acknowledged before, we purposefully decided to include a wide variety of approaches; therefore, articles that used both internal and external coaching approaches were considered. Moreover, coaching strategies outside and including the Hackman and Wageman [9] model were included. Notwithstanding, the wide range of research methods and perspectives made it difficult to capture all competencies and intricacies pointed out by the three research streams. For this reason, all four coders were instructed to focus on extracting information from the articles that explicitly focused on *how* to actually increase/improve each of the three areas, rather than focusing on general performance improvements. For example, if an article noted a training resulted in “improvements in group motivation” but did not explicitly address how this was achieved, that was noted but not included for our thematic analysis. On the contrary, if the article noted “improvements in group motivation” and specified this was achieved via goal setting, this was included for our thematic analysis. All coders took notes on each article and then the coders met to consolidate the notes (see Appendix A). Two coders did the first half, and the two coders did the last half (all articles were double-coded). From this, the two leading authors discussed commonalities among all articles and extracted the nine competencies presented below.

## 4. Overview of Internal Team Coach Competencies

The systematic review revealed a variety of methods through which leaders and teams are being trained to improve the three specific processes team coaching has been known to improve. Overall, the literature yielded a total of nine main competencies internal team coaches need to have to properly coach their teams, with three under each category. While the majority of these have been previously highlighted by the literature (e.g., initiating structure) [24], it is their combination that makes an internal team coach separate from simply being a team leader—as not only are they leading their team into success, but purposefully ingraining in their team members how to complete tasks autonomously and self-regulate (e.g., allows autonomy, reflective practice). Below, we describe each core category of competencies, alongside detailing each competency, followed by a general explanation on how to train these behaviors. Table 2 provides a summary of these competencies and their corresponding area.

## 5. Competency Area #1

### 5.1. Increasing Effort and Motivation

A *team* is a group of two or more people who interact with each other, rely on each other, share a common purpose, and view themselves as a unit [25]. A key part of distinguishing a leader who has employees working for them vs. a leader who is leading a team is an underlying common purpose—a *shared goal*. Organizational psychologists understand motivation as a goal-striving process that involves direction, intensity, and persistence in activities to achieve said goal [26]. At the team level, this becomes increasingly complicated, as an internal team leader must be able to articulate what the team’s goal is and how the team is going to achieve it. However—, an internal team leader who wants to engage in team coaching must go beyond establishing a goal or delineating who does what. An internal team coach will engage in three key behaviors: initiating structure, allowing autonomy, and offering continuous support for the team’s goal completion. In other words, an internal team coach does not simply lead, but guide, and moreover—allows for others to guide themselves.

#### 5.1.1. Initiates Structure

Organizational psychology has long established that effective leaders initiate structure, dating back to path-goal theory [24]. An effective leader will initiate psychological structure by engaging in behaviors such as specifying “procedures to be followed, clarifying […] expectations of subordinates, and scheduling work to be done” [24] (p. 321). In reviewing the literature, it became clear that leader effectiveness (e.g., [27,28]) and team effectiveness (e.g., [29,30]) hinge on setting clear goals, necessary to keep team members motivated. The team coaching literature reiterated this (e.g., [31,32])—albeit making it abundantly clear that, to even begin guiding one’s team, an internal team coach needs to engage in initiating structure via goal setting. However, the literature revealed a variety of ways in which a team coach can do this—going beyond simple goal setting and highlighting methods such as team chartering (e.g., [33]) and continuously clarifying aims (e.g., [34]) and concerns (e.g., [35]). Therefore, an effective internal team coach will not just set a goal but agree on it with their team, clarify expectations around team behavior and the mission at hand, and make an effort to ensure everyone is caught up on the mission as it progresses. Notwithstanding, the way of delivery in establishing and reinforcing structure matters, marking the need for allowing autonomy amidst a team mission.

#### 5.1.2. Allows Autonomy

An internal team coach allows autonomy in how team members connect to the team’s mission and allows for flexibility in their work process. By doing so, the internal team coach can help team members take responsibility for their progress. All three streams of literature reiterated the importance of autonomy/flexibility for effective performance. For example, one leadership intervention found that facilitating autonomy and other psychological needs positively impacted work engagement and performance up to two months post-training [36]. In virtual teams, working autonomously was also identified as a critical competency [37]. In healthcare teams, self-management allowed for empowerment and motivation within the team [38]. The general coaching literature noted that coaching can help individuals become autonomous in their learning processes [39]. Altogether, these findings align with self-determination theory (SDT), which holds autonomy as necessary for motivation [40]. As another example, in an external team coaching approach, Körner et al. [31] noted the importance of emphasizing *responsibility* for team coaching to be effective down the line, as this allows people to understand and accept the process. These approaches suggest that allowing flexibility and choices in how team members complete their tasks will motivate members (see [41]). Therefore, an internal team coach must not only initiate structure but also allow autonomy within said structure and facilitate team member connections to the goal by allowing everyone to “bring a piece of the pie” to the table. This is a key boundary between leading and team coaching: not simply delegating tasks but helping others learn how to approach them.

#### 5.1.3. Offers Support

In one external team coaching approach, Woodhead [42] found that the coach supported the team in maintaining dialogue, completing their goal, maintaining a safe environment, and decision making. Another team coaching approach involving a discussion of goals, consideration of consequences, and evaluation suggests team coaching could support goal attainment for a team [35]. In the leadership training literature, leadership support was positively associated with team functioning [43]. The team training literature noted mutual support as imperative for handling positive and negative situations [44]. Hence, another theme emerged: the importance of a team coach’s support throughout the goal-attainment process. However—the type of support offered should be focused on the task rather than on interpersonal relationships.

Hackman and Wageman’s [9] original team coaching approach focused on the team task rather than interpersonal relationships, deemed a “structural view”. This structural perspective argues that conflict arises due to higher-order process issues, like an unclear team goal, rather than interpersonal differences (see [2]). This description does not negate the existence or importance of interpersonal relationships within a team but instead argues that if a team focuses on the task and is clear on what needs to be done, interpersonal relationships will facilitate rather than cloud goal attainment. In keeping with this original definition, and with recent work on team dynamics finding *task cohesion* is more predictive of team performance compared to *social cohesion* (see [45])—internal leaders who seek to implement team coaching behaviors should offer support for the completion of the task, attempting to clarify steps around the goal to avoid conflict. One exciting way to do this as an internal team coach is to have a vision for the team. For example, Aripin et al. [46] described a successful leader as someone who explains, directs, and demonstrates the organization’s goal and vision, inspiring employees and providing them with purpose and direction. The ability of a leader to think creatively and come up with new concepts, alongside having the ability to combine this with strategy and goal setting, can be conducive to organizational development [47]. Therefore, an internal team coach can support the team’s mission by helping members visualize the “end game”, offering expertise when needed—and not as an imposition on other’s roles. In other words, the internal team coach must guide others in the right direction, not force them down that path.

## 6. Competency Area #2

### 6.1. Improving Interpersonal Processes

Many researchers in the team coaching field believe that prioritizing interpersonal relationships within a team can yield positive results (e.g., [18]). Nevertheless, as stated above, a structural view of team coaching necessitates focusing on the task rather than team member relationships. However, whether a focus on relationships is helpful or not is not a debate endemic to the team coaching field, as others have also long argued over the effects of interpersonal conflict on performance, with some finding it beneficial (e.g., [48]) and some not (e.g., [49]). In this section, we discuss how internal team leaders can stay focused on the task while fostering positive interpersonal relationships and psychological safety. We do this not from a psychodynamic perspective but from a task perspective, where teams can learn to self-regulate by fostering a safe environment. This process begins by being aware of the team’s needs, maintaining interpersonal sensitivity, and monitoring teamwork behaviors over time.

#### 6.1.1. Acknowledgement

Recent literature has emphasized the importance of leader well-being for follower performance (e.g., [50]). Dalgaard et al. [51] recognized this and developed a training program for managers to improve their well-being, arguing that by doing so, leaders’ ability to promote staff care also improves. Though exhibited in different ways, this awareness of one’s own needs and limitations repeatedly came up in the literature not only as an essential precursor to overall performance but also effective teamwork (e.g., [52]) and effective leadership (e.g., [53,54]). Therefore, an internal leader must know their needs and limitations as a primary step before engaging in team coaching. This relates to one’s needs and limitations about the team goal. For example, if the goal is to build a rocket, and the team leader is not an engineer, a leader who engages in team coaching would be aware of their limitations, establish boundaries, and use this self-awareness to empower the team in other ways. They would give autonomy to the engineer in the team to lead under certain circumstances and empower them to speak up when necessary.

Moreover, an effective internal team coach must also develop awareness and understanding of the team’s needs. This can be achieved by assessing team member styles (see [32]) and understanding a team’s reward preferences (see [55]). For example, while a leader might reward their team for making progress by hosting a pizza party, a leader who engages in team coaching is aware of what team members want as a reward and will gauge whether this would be in line with the team’s desires. This is because an internal team coach maintains an open dialogue with their team and engages in goal setting with them, not *for* them. As another example, if a deadline is impossible given organizational constraints, an effective internal team coach will engage with their team to attempt to remove barriers rather than simply leaving them there [10]—also tying to the prior competency of *offering support.*

#### 6.1.2. Maintains Interpersonal Sensitivity

Though a focus on interpersonal relationships should not be the goal of an internal team coach, they must maintain interpersonal sensitivity to enable effective performance. Sensitivity came up as a critical competency in multiple streams of research (e.g., [47,56,57]). Therefore, an internal team coach should exhibit and model norms of respect to all team members, regardless of who they are. Nevertheless, it is essential to note that engaging with the competency of interpersonal sensitivity is much more effective when woven into a team’s norms (see [58]). A leader can demonstrate interpersonal sensitivity in various ways, from establishing rapport [57] to simply asking team members how they feel about where they are at (e.g., [32]). By asking questions and remaining aware of who needs what, as well as offering support when needed (see *offering support* competency), an internal team leader can maintain interpersonal sensitivity that, while not actively playing a role in the completion of the task, facilitates it by creating a psychologically safe environment (see [11]).

#### 6.1.3. Engage in Open Dialogue

One of the main strengths of internal team coaching is that a team will learn how to self-regulate rather than rely on external interventions or training to engage in effective team processes [2]. To accomplish such a degree of self-regulation, an internal team coach must maintain open lines of communication with their team. In opening goal setting and norm-setting as a team process rather than a strict leader process, team members develop their own relation to the team task. For example, in an external coaching intervention, Woodhead [42] found that, eventually, team members developed a deep sense of trust with each other that enabled collaborative decision making and allowed the team to put the interests of the team first (as opposed to their individual needs). While in this case, an external actor enabled this, internal team coaches can also do this in their teams (see [59] for a combined approach), as evidenced by the team literature that found that inclusive communication enabled a shared sense of responsibility for the team’s performance [38]. Moreover, the team and leadership training literature suggest that open communication can be trained (e.g., [44]) and promoted [60].

## 7. Competency Area #3

### 7.1. Increasing Knowledge and Learning

For a team to accomplish any goal, they must be able to share and integrate knowledge. Long-standing research in team science indicates that this is achieved via team cognition, the team system’s overlapping representations, the team’s interdependent tasks, the roles and responsibilities of each team member, and their shared experiences [61]. Part of team cognition is a team’s *shared mental model*, defined as knowledge structures that allow teams to have a shared understanding of the task at hand. For an internal team coach to aid and influence team knowledge and learning, they must understand how team cognition and shared mental models develop over time. Moreover, they must facilitate their growth and aid the team in reflecting during goal and after goal completion. This section describes what an internal team leader must understand to facilitate these processes.

#### 7.1.1. Organizes Team Information

A challenge for teams is adjusting to changing demands and conditions. One reason this is challenging is that, in a team, it can be difficult to discern who knows what. A critical competency a team leader needs to have to engage in internal team coaching is the ability to organize team information. One way to organize team information is via a shared mental model that an internal team leader can facilitate via information sharing and exchange [61]. An internal team leader should promote information sharing, helping embed psychological safety into the team’s culture (see [41]). This is even more important for teams in turbulent and complex environments, where teams heavily rely on their shared mental model and team leader to function effectively [33]. This is not a stand-alone competency; it is heavily dependent on initiating structure and properly establishing team roles. However, as opposed to traditional ways of leading, team leaders who seek to engage in team coaching should allow people to get involved in the process of knowledge sharing (see [32]). Internal team coaches see this process as collaborative and not as an imperative. They help organize the team, but they also rely on their team. External coaching approaches have established the importance of self-understanding and expanding awareness (see [62]), and the team training literature suggests that, when people know and understand what everyone is contributing to the overarching goal, they will support other’s contributions [63]. Therefore, while it is not the job of a leader to know everything, it is their job to make sure team members know who to ask and organize this process.

The team coach can organize team members’ expertise, connecting team members strategically [10]. For example, the leader can make a specific attempt to connect the engineer with the team scientist to execute the launch of a product, helping them form a working relationship. Moreover, by being purposeful about initiating structure, team coaches can even align certain team member expertise and potentially even better synchronize their efforts when they work together (see [35]). Team members who are better at integrating knowledge will learn from each other and gain new knowledge and perspectives of the team and the complex problems they are solving (see [64]). In essence, what will generate new knowledge and subsequent learning will be the team coach’s ability to encapsulate the team’s knowledge and disseminate it to the appropriate team members who are working on a component of the overarching team tasks.

#### 7.1.2. Facilitates Information Sharing

Psychological safety—a reiterated theme in all three streams of literature (e.g., [65,66,67])—is vital for knowledge creation because it allows people to share information without fear of reprimand (see [11]). Hence, a leader who wants to engage in team coaching and increase team knowledge and learning must facilitate knowledge sharing by cultivating a positive environment. This begins by playing into the interpersonal competencies discussed above. Moreover, other extant literature suggests fostering psychological safety and a positive environment by being willing to listen to different team members’ perspectives on different issues [58]. Creating a positive environment is not a one-and-done process. Effective teams will take the opportunity to learn from their mistakes and engage in reflective practices [58], making sure the team understands what happened, what could be done better, and what adjustments will be made, as discussed below.

#### 7.1.3. Reflective Practices

Active learning is when a leader structures how the team learns and refines knowledge. The team training and simulation literature finds that when teams set time aside to reflect on their plans and how everyone contributed to the goal, knowledge can increase (see [63,68]). Moreover, leaders who engage with learning as a continuous process could potentially continue to enhance knowledge. For example, Seeg et al. [69] found that a post-training intervention six weeks after training enhanced knowledge transfer. Internal team coaches can learn from this, remind their teams of past experiences, and apply this knowledge to future challenges rather than starting with a slate anew every time something comes up.

Team reflection and planning are critical for laying the groundwork for team functioning [35,70]. Internal team coaches engage in active learning and can do so via feedback and debriefing. This can be done via feedback and team debriefing to promote knowledge integration and learning (see [41]). When a team coach provides constructive feedback on teamwork, they can create a culture in which information is freely exchanged, and feedback is viewed as an opportunity for learning. Therefore, when mistakes or failures occur, the team coach can use feedback and debriefing to learn for the team and correct mistakes. The effectiveness of the feedback will depend on the tone after the teamwork breakdown and how the team coach presents the information to the team—necessitating the above critical competency of interpersonal sensitivity.

## 8. Training the Competencies

All three streams of literature contained a plethora of ways in which leaders, coaches, and teams can improve their dynamics. However, one idea was so widely reiterated that it is essential for organizations to understand one crucial step prior to implementing internal team coaching as a strategy. Leaders must conduct a needs analysis before engaging in team coaching (see [31]). Whether organizations choose to implement a formal training workshop detailing how to perform one or whether internal leaders decide to undertake this practice themselves, both approaches must delineate the importance of understanding what the goal is, where the problems lie, and what can be performed to approach the goal and surrounding issues. Other than this, all three streams of literature contained a variety of ways to train these specific behaviors. However, a particularly strong sub-stream of literature was the simulation literature that emphasizes *learning by doing.*

Leaders who want to engage in internal team coaching must practice by doing. They must involve their teams in goal setting, norm-setting, feedback sessions, debriefs, and so forth. Organizations who want to engage in more formal internal team coach training should teach team coaches how to set specific goals, give effective feedback (e.g., give task-centered feedback and make it a two-way street) [71], and conduct effective debriefs. They can teach these behaviors by strategies such as role-playing and a varying degree of simulation techniques (see [72])—as what matters is not the perfect replication of physical environments or situations [73], but allowing for practice prior to a real-time engagement. A promising point of simulation-based training (SBT) techniques is their accessibility. Training does not have to be an expensive endeavor. Organizations simply have to allow leaders a space in which they can practice these techniques and learn about these competencies. Nevertheless, it is important to note that team coaching is an approach, and not the approach. Different team developmental interventions (TDIs, see [20]) can also be helpful, depending on the issue that an organization is facing. Team coaching can be both a formal training intervention and a process intervention where an individual simply chooses to engage in certain behaviors [2]. However, depending on what the end-goal is, it is possible that team training interventions are better suited, or if a leader is not dealing with an intact team, a leadership intervention might fit best. All-in-all, team coaching is a best fit for a leader that leads a compact team with a specific goal, and the evaluation of needs is necessary prior to any intervention.

## 9. Limitations

The competencies put forward by this review assume that research from the leadership and team training sphere translate into the team coaching sphere as a leader-behavioral approach. This is a strong limitation of the research put forward—but given the lack of empirical evidence in the team coaching sphere, we believe it was necessary to keep moving forward. Given that many internal team coaches do not undergo formal training, putting together a list of key competencies allows the literature to understand what behaviors leaders need to exhibit to call themselves team coaches. However, a much-needed pathway for future research given the assumption of this paper is to develop a scale that evaluates these behaviors in internal team coaches to attest to their existence. Moreover, such a study should measure these competencies in team coaches and most importantly—note whether they correspond to each of the three areas noted here. It is important that these competencies build on each other and lead to an improvement in team performance, and for this, empirical research with control groups is necessary. By doing so, research can attest to the existence of these competencies empirically.

Another important limitation of this review is the generalizability of our findings. Given that these competencies were extracted from a variety of literature streams, we are confident that they are applicable across a variety of settings where the task is clearly defined. However, it should be noted that there has been very little research on understanding where team coaching is useful. The few empirical studies that have been conducted on team coaching have been focused on the healthcare and innovation sphere, which are two areas in which the task tends to be clearly defined. Therefore, more insights are needed on whether these competencies (and team coaching at large) are useful in other organizational settings, such as corporate settings.

## 10. Future Directions

The need for more empirical research to solidify the team coaching approach has been previously remarked [2] yet needs to be reiterated. However, advancements are being made, and more recent research has attempted to be more descriptive and transparent in their approaches (e.g., [31]). To continue streamlining the literature, researchers should make their approach transparent (i.e., describing whether they are taking a task-focused or psychodynamic perspective, etc.). Moreover, in reviewing the team coaching literature, few authors discussed why they undertook an internal vs. an external team coaching approach. We encourage authors to include this debate in their research, which can help organizations understand when one might be needed over the other. As brought up by the reviewers of this paper, an especially useful empirical lens would be to undertake a comparative study that delineates under what circumstances internal coaches work and under which conditions external coaches are needed.

On top of streamlining the literature via the two points discussed above, there are a variety of ways in which researchers can continue to strengthen team coaching via empirical research. Given that team coaching is a task-oriented approach meant for teams with a specific task in mind and who are more compact, it would be interesting to explore whether this method of leadership works for virtual teams or fluid teams. On top of this, and as remarked by this paper’s reviewers, there has been a complete dearth of research focusing on team coaching across cultures, with some exceptions (e.g., [74]). Therefore, a clear research need involves understanding how these competencies show up across cultures, such as “initiating structure” and potential differences among vertical vs. horizontally organized cultures.

## 11. Conclusions

This paper presents a unique perspective on team coaching, viewing it as an internal leadership approach. Individuals can effectively lead their teams by establishing a structure that allows autonomy and support, maintaining interpersonal sensitivity, and fostering open dialogue. Additionally, we propose that internal team coaches can enhance team performance by organizing information, facilitating information sharing, and embracing reflective practices as part of a continuous learning process.

This paper delves into the practical aspects of team coaching, aiming to identify the key competencies, behaviors, and actions that internal team leaders can adopt to enhance team dynamics. Building on previous research that has established the effectiveness of team coaching in improving performance, this paper provides a comprehensive list of strategies for motivating one’s team, managing interpersonal processes within it, and fostering team knowledge and learning.

## Figures and Tables

**Figure 1 behavsci-14-00452-f001:**
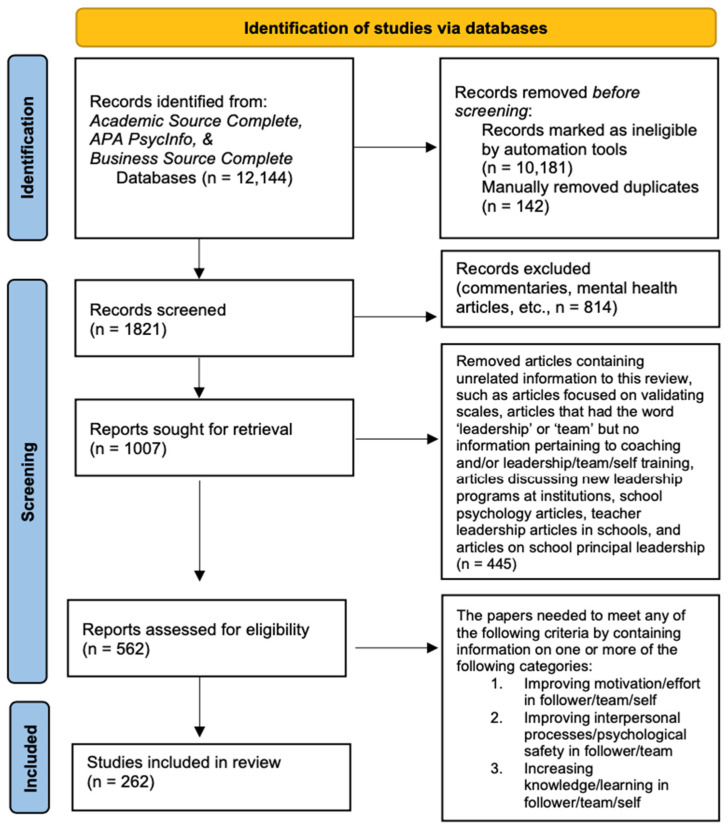
Systematic review flowchart.

**Table 1 behavsci-14-00452-t001:** Team coaching terminology.

Terminology	Definition
Team coaching	A task-focused leadership process approach that helps team leaders accomplish a shared team goal by increasing group effort and motivation, improving interpersonal processes, and increasing team knowledge and learning [2].
Internal team leader	A leader of a team with a shared goal. Anyone who leads a team is an internal team leader, as opposed to an external coaching consultant brought in to help a team. Example: A surgeon leading a surgical team.
Internal team coach	An internal team leader leading a team with a shared goal implementing strategies and actively using team coaching techniques, such as goal-setting, team norming, and active learning. Example: A surgeon leading a surgical team using team coaching as their leadership strategy.
External team coach	An external consultant that uses team coaching as their main developmental tool to help a team advance towards their shared goal. Example: An external consultant coaching a surgical team to improve their team performance.

**Table 2 behavsci-14-00452-t002:** The nine internal team leader competencies.

Increasing Effort and Motivation	Improving Interpersonal Processes	Increasing Knowledge and Learning
*Initiates structure* by setting team goals, team chartering, and clarifying aims.	*Acknowledges and is aware* of their own needs and that of the team.	*Organizes team information* by establishing a shared mental model.
*Allows autonomy* in team member work while committing to team goals.	*Maintains interpersonal sensitivity.*	*Facilitates information sharing* by cultivating a positive environment.
*Offers support* for team goal completion.	*Engages in open dialogue* with their team.	Engages in *reflective practice* by debriefing and providing feedback to the team.

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
