# Peer review of "How to Make an Internal Team Coach: An Integration of Research"

_behavsci, 2024, doi:10.3390/bs14060452_

Round 1

Reviewer 1 Report

Comments and Suggestions for Authors

Author Response

Thank you for your time and consideration in reviewing our manuscript. We have submitted a PDF that addresses every comment received.

Reviewer 2 Report

Comments and Suggestions for Authors

Thank you for submitting your manuscript, "How to Make an Internal Team Coach: An Integration of Training Research." I greatly appreciate this work and find it to be a timely topic. Building on the paper's strengths, I would also suggest considering the following areas for improvement.

1. The need for the study is well-presented. However, the theoretical underpinnings of the research idea could benefit from further elaboration. I encourage the authors to strengthen the theoretical section by drawing more heavily on existing theories from the literature. While the theory of team coaching has been mentioned, what additional theoretical frameworks could be explored?

2. Following up on my previous suggestion, I believe it would be valuable to provide a framework for understanding what constitutes competence in the field of team intervention. This framework would be particularly helpful in better understanding of increasing effort and motivation, improving interpersonal processes, and increasing knowledge and learning as core categories of competencies. 

3. For future research directions, it would be interesting to explore the application of team coaching to other types of teams, such as those with multiple memberships, fluid structures, or virtual settings. These teams present unique challenges and require specific facilitation approaches.

Author Response

Thank you for your time and consideration in reviewing our manuscript. We have submitted a PDF document for Reviewer #2 addressing all comments brought up.
